# Cancer Stem Cells in Oropharyngeal Cancer

**DOI:** 10.3390/cancers13153878

**Published:** 2021-08-02

**Authors:** Mehmet Gunduz, Esra Gunduz, Shunji Tamagawa, Keisuke Enomoto, Muneki Hotomi

**Affiliations:** 1Department of Otorhinolaryngology and Head and Neck Surgery, Shinmatsudo Central Hospital, Shinmatsudo 1-380, Matsudoshi 270-0034, Chiba, Japan; mgunduz@ims.gr.jp; 2Department of Otorhinolaryngology and Head and Neck Surgery, Wakayama Medical University, Kimiidera 811-1, Wakayamashi 641-8509, Wakayama, Japan; tamashun@wakayama-med.ac.jp (S.T.); kenomoto@wakayama-med.ac.jp (K.E.); 3East Clinic Moriya Keiyu Corporation, Matsunami 1630-1, Moriya 302-0108, Ibaraki, Japan; gunduz.e@keiyu.or.jp

**Keywords:** oropharyngeal cancer (OPC), human papillomavirus (HPV), cancer stem cells (CSCs), CSC markers, prognosis, tumor heterogeneity

## Abstract

**Simple Summary:**

Although there has been improvement in our understanding about cancer stem cells recently, we still don’t know much about cancer stem cells of oropharyngeal cancer. Lack of knowledge solely on oropharyngeal cancer together with the information of human papilloma virus status, which is a specific factor of prognosis in oropharyngeal cancer, hardens to elucidate the distinction of the underlying mechanisms of cancer stem cell behavior. To proceed to an effective and durable therapy in oropharyngeal cancer it is necessary to reveal cancer stem cell function and related factors like its plasticity, niche, and pathways. Therefore in this review we aimed to contribute to this emerging area by focusing on the current literature and future prospects.

**Abstract:**

Oropharyngeal cancer (OPC), which is a common type of head and neck squamous cell carcinoma (HNSCC), is associated with tobacco and alcohol use, and human papillomavirus (HPV) infection. Underlying mechanisms and as a result prognosis of the HPV-positive and HPV-negative OPC patients are different. Like stem cells, the ability of self-renewal and differentiate, cancer stem cells (CSCs) have roles in tumor invasion, metastasis, drug resistance, and recurrence after therapy. Research revealed their roles to some extent in all of these processes but there are still many unresolved points to connect to CSC-targeted therapy. In this review, we will focus on what we currently know about CSCs of OPC and limitations of our current knowledge. We will present perspectives that will broaden our understanding and recent literature which may connect to therapy.

## 1. Introduction

Head and neck cancer is the sixth most common malignancy worldwide and includes tumors from the oral cavity, pharynx, larynx, nasal cavity, paranasal sinuses, thyroid, and major as well as minor salivary glands [1] The thyroid, nasal cavity as well as paranasal sinuses and salivary gland tumors are usually considered to be a different group and when the head and neck cancer is mentioned, mostly squamous cell carcinomas deriving from oral cavity, pharyngeal and laryngeal mucosa are taken into consideration. Most common risks for head and neck cancer are smoking, heavy drinking and virus contamination [2]. Within these groups, oral cavity and especially oropharyngeal cancer are related with human papilloma virus (HPV) infection. The identification of HPV especially in the USA and European countries in oropharyngeal cancer resulted in increased basic and clinical research and better outcomes for diagnosis and treatment of this head and neck cancer type.

Oropharyngeal cancer (OPC), a common subtype of head and neck squamous cell carcinoma (HNSCC), is mainly associated with tobacco as well as alcohol use and, HPV infection. Thus, oropharyngeal cancer has recently been categorized into HPV associated (positive) or unassociated (negative) subtypes, which show quite different etiologic as well as genetic backgrounds and therapeutic outcomes. Although high risk HPV subtypes are HPV16, 18, 31, 33, 35, 45, 51, 52, and 56 in relation to causing cancer of the head and neck, cervix, anus, vagina, vulva, and penis, HPV16 is the most common type in head and neck cancer [3]. Mechanism of HPV integration to the host genome is not clear yet but fusions through break-points of cellular and viral genome or the amplified segments of a genomic sequence flanked with HPV genome which is also found in patient samples as focal copy number elevation at sites of HPV integration are mainly suggested mechanisms [4]. While HPV-positive oropharyngeal cancer has low or no common genetic abnormalities, such as p53 mutation, and is directly associated with contamination of high risk HPV subtypes, HPV-negative oropharyngeal cancer is closely related with smoking and excessive alcohol consumption and demonstrates commonly activated mutation of oncogenes such as EGFR, RAS, PI3 kinase or functional loss of tumor suppressor genes such as p53, p16, and RB1 or both [2]. Thus, underlying mechanisms and as a result prognosis of the HPV-positive and HPV-negative OPC patients are quite different (Table 1). There are also conflicting reports about HPV status and HNSCC which may be due to not screening cancer of the oropharynx and the anterior oral cavity separately or different sampling techniques such as saliva, biopsies, and brushing and methods used to detect HPV status from those samples through polymerase chain reaction (PCR), dot-blot hybridization, and Southern blotting [5].

Cancer stem cells (CSCs) are a subgroup of cells in the heterogeneous tumor bulk, which have the ability of self-renewal and differentiation, like stem cells. They are supposed to have a role in tumor invasion, metastasis, drug resistance, and recurrence after therapy and therefore are accepted as one of the emerging targets for cancer therapy. Recently they have been studied by various researchers throughout the world and their roles in carcinogenesis, tumor invasion, metastasis, drug resistance, and recurrence after therapy have been shown but still more research and evidence are necessary to move forward to CSC-targeted therapy [7,8]. To be able to comprehend our knowledge of the CSC of OPC, in this study we will first present CSC origin and model followed by CSC markers of HNSCC. Then we will review the recent literature and our experience about CSC of OPC, and limitations of our current knowledge. We will discuss different perspectives, which may connect to better diagnostic as well as prognostic and therapeutic options. Lastly, we will comment on for further investigations that can be performed to connect CSC-targeted therapy for OPC.

### CSC Model and Origin

CSCs were first identified in acute myeloid leukemia in 1997. The cells with CD34^++^ CD38^−^ cell-surface antigen were only 0.2% in the tumor but had the potential to form neoplasms in the immune-deficient mice. Conversely, even though CD34^+^ CD38^+^ cell-surface antigen cells were highly detected in the tumor they couldn’t engraft new neoplasms [9]. In 2007, CSCs of head and neck squamous cell carcinoma were first identified. Prince et al. defined CD44^+^ cells comprising of lower than the 10% of HNSCC cells but could give rise to new tumors and the new tumors formed from CD44^+^ purified cells could reproduce tumor heterogeneity and could be serially passaged like stem cells [10].

The hierarchic model of cancer which is also known as the CSC model implicates that only specific cells have the ability to form cancer cells. Although it is still unclear which cancers or which cancer stem cells expressing specific markers follow this model, increasing evidence supports this hypothesis. Contrary to the stochastic model of tumor growth in which all the tumor cells stochastically have the potential to self-renewal and differentiate, in CSC model CSCs are responsible for causing different lineages in the tumor that leads to tumor heterogeneity [11,12] (Figure 1).

The origin of CSCs is still under discussion. Three possibilities are raised that CSCs can be formed from: stem cells, progenitor cells or differentiated cells [13]. However, it is also possible that even some of differentiated cancer cells may gain CSC properties through oncogenic pathways and environmental stimuli. Therefore, it is questioned whether CSCs differentiate in a unidirectional hierarchic way like in the CSC model. Moreover, evidence supports that cancer cells have plasticity and can acquire CSC properties [14,15].

## 2. CSC Markers of HNSCC

In this part we will summarize widely used CSC markers in HNSCC studies (Table 2).

### 2.1. ALDH1A1

Aldehyde dehydrogenase 1 (ALDH1A1) which is in the major pathway of alcohol metabolism is located at chromosome 9q21.13 with 13 exons [16]. ALDH1A1 is a valuable prognostic CSC marker. Chen et al. defined ALDH1A1 as a CSC marker in HNSCC, which was previously used as CSC marker in various cancers. They showed that ALDH1^+^ cells displayed resistance to radiotherapy and had ability of generating tumors [17]. Likewise it was reported that ALDH1A1 is a highly selective marker for CSCs in HNSCC [18]. Qian et al. analyzed HNSCC specimens of which 80% was oropharyngeal cancer for ALDH1A1 expression and its relation to prognosis. Their results showed that HNSCC patients with ALDH1A1 expression displayed a significant *p* value (*p* = 0.011) for poor prognosis and those of oropharyngeal cancers with ALDH1A1 expression showed worse prognosis (*p* = 0.001) [19]. Similarly Szafarowski et al. compared CSC markers of HNSCC and their results revealed that ALDH1A1^+^ patients showed 5.25 times worse overall survival (OS) than ALDH1A1^−^ patients (*p* = 0.01) and only ALDH1A1 positivity had a significant effect on OS of HNSCC patients (*p* = 0.02) compared to other CSC markers of CD44, CD24 and CD133 [20]. In another study, it was confirmed that patients with ALDH1A1^+^ had worse prognosis but also concluded that ALDH1A1 and CD44, alone or together, was not enough to identify CSC subpopulations [21]. Contradictory to this data we previously characterized CSCs of OPC and had been successful to isolate CSCs by ALDH1A1 marker and CSCs have the ability to form tumor spheres [22]. ALDH1A1 is one of the most specific markers that are used for HNSCC CSC research [23,24,25,26].

### 2.2. CD44

CD44 is a cell-surface glycoprotein involved in cell-cell interactions, cell adhesion and migration, which is located at chromosome 11p13 with 21 exons [27]. First defined by Prince et al., CD44 has been frequently used as a CSC marker in various HNSCC studies [10,28,29,30] and has been shown to play important role in HNSCC cancer stemness [31,32,33]. CD44 was also found to be related to angiogenesis, tumor aggressiveness, and worse prognosis in HNSCC [34,35,36]. In addition, a meta-analysis study displayed worse prognosis in pharyngeal and laryngeal cancer with CD44 expression but not in oral cancer [37].

### 2.3. BMI1

BMI1 is a proto-oncogene located at chromosome 10p12.2 with 10 exons. It is a member of polycomb group complex 1 (PRC1) which is an epigenetic repressor of regulatory genes in embryonic development and self-renewal of somatic stem cells via chromatin remodeling [38]. It was shown that inhibiting BMI1 sensitized tumors to cisplatin and eliminated lymph node metastasis in vivo, in vitro and primary human HNSCC samples contained highly tumorigenic, invasive, and cisplatin-resistant BMI1^+^ CSCs [39,40]. Tumor growth was also suppressed by inhibiting BMI1 pharmacologically in HNSCC and targeting BMI1 related CSC in oral squamous cell carcinoma (OSCC) has been shown as a clinically relevant anticancer therapy [41,42].

### 2.4. OCT4

OCT4 is located at chromosome 6p21.33 with six exons. It encodes a transcription factor that plays role in embryonic development and stem cell pluripotency [43]. It was reported as a CSC marker in HNSCC [32,44]. OCT4 was found to regulate epithelial-mesenchymal transition (EMT) in OSCC [45]. Because of its relation to poor prognosis it can be used as a predictive prognostic marker of HNSCC [46].

### 2.5. SOX2

SRY-box transcription factor 2 (SOX2) is located at chromosome 3q26.33 and it has no intron. It plays a role in the regulation of embryonic development and the determination of cell fate [47]. It has been shown to regulate CSC of HNSCC [48]. There are conflicting reports about its high expression related to prognosis [46,49,50].

### 2.6. CD133

CD133 is located at chromosome 4p15.32 with 35 exons. It is a transmembrane glycoprotein expressed on adult stem cells. It is supposed to suppress differentiation to maintain stem cell properties [51]. CD133 high expression was shown to increase cancer stemness and cause cell cycle arrest in HNSCC cell line resulting in chemoresistance [52]. Chen et al. proposed CD133/Src axis might be a potential therapeutic target in HNSCC because of being a regulatory switch to gain of EMT and of stemness properties in HNSCC [53]. It is also found to be a biomarker and predictor of prognosis [54].

## 3. OPC CSC Pathways

CSC markers ALDH1A1, CD44, BMI1, OCT4, SOX2, and CD133 and their effects on cancer stemness, metastasis, prognosis, chemo/radiotherapy resistance and recurrence have been studied in HNSCC by our group and other researchers [10,17,22,31,32,33,34,35,36,39,40,41,42,45,46,48,49,50,52,53,54]. Although there are conflicting reports about their expression and as being a prognostic marker in HNSCC, the differences may be due to factors such as use of cell lines in vitro vs primary tumor samples in vivo, the used isolation techniques such as fluorescence activated cell sorting (FACS) vs magnetic beads activated cell sorting (MACS), tumor sample/cell line kind e.g., pharyngeal cancer vs. oral cancer vs. laryngeal cancer, patient or sample size and, if primary tumors was used before chemo/radiotherapy vs after chemo/radiotherapy. Additionally, intratumor heterogeneity may also reflect different results.

CSC studies involving solely OPC are very limited. Moreover, underlying mechanisms are different due to the HPV status. Rietbergen et al. analyzed 711 oropharyngeal squamous cell carcinoma (OPSCC) patients from two Dutch university hospitals and showed that HPV-positive patients had lower CD44 and CD98 expression than HPV-negative patients. Moreover HPV-positive patients with high CD98 expression showed significantly worse overall survival (OS) and progression-free survival (PFS) rates compared to patients with low percentage of CD98 cells [55]. Likewise Näsman et al. presented that HPV-positive patients with CD44 absent/weak expression displayed significant favorable 3-year disease-free survival (DFS) and overall survival (OS) [36]. In a study, OPC patients who had undergone radiation therapy, it was shown that CD44 negative patients had significantly higher PFS and locoregional control (LRC) than CD44 positive patients. Furthermore, p16 protein positive (likely to be HPV-positive) and CD44 negative patients showed the best LRC while p16 protein negative (likely to be HPV-negative) and CD44 positive patients had the worst LCR [56]. These results indicate CD44 expression is low in HPV-positive cases while it is high in HPV-negative cases and if CD44 expression is high in HPV-positive cases, it results in a worse outcome. Our group isolated CSCs from HPV-negative cell line of UT-SCC 60A by CSC marker ALDH1A1 and showed that CSCs formed tumor spheres. We also detected significantly high expression of OCT4, SOX2, KLF4 and BMI1 in the HPV-negative OPC CSCs as compared to the cancer cells, while CD133 expression was not different in the CSCs and the cancer cells. Those of the CSCs showed resistance to cisplatin treatment [22]. BMI1 was also found to be expressed more in HPV-negative OPC than HPV-positive OPC [57].

Orai1 was shown to be regulator of CSC phenotype by Lee et al. in oral/oropharyngeal cancer. According to their data Orai1 has been highly expressed in oral/oropharyngeal cancer and activates downstream molecule NFATc3 which proposes Orai1/NFAT axis to have importance on CSC in OPC [58]. In a later study Lee et al. introduced NFATc3 as a critical factor which affects cancer stemness through NFATc3-OCT4 axis in oral/oropharyngeal cancer. Their data included NFATc3 was highly expressed in CSC and required for self-renewal of CSC. Furthermore, their data indicated not only the gain of CSC phenotype but also gain of ALDH1A1^+^ high cell population, morbidity and drug resistance when NFATc3 was ectopically expressed in immortalized oral epithelial cells as well [59].

Interestingly, Hufbauer et al. showed that HPV16 targets migratory and stationary stem cells and aberrantly expressed miR-3194-5p and miR-1281 in migratory CSCs, which might be the reason of OPC progression and metastasis [60]. Finding HPV16-positive HNSCC to have more CSC than HPV16-negative HNSCC, Zhang et al. discussed that rather than amount of CSC, CSC phenotype may be more important in the therapy resistance [61].

## 4. From the View of CSC Research to the Therapy of OPC

Possessing the ability of self-renewal and differentiate, CSCs are considered to be one of the emerging targets for cancer therapy. They are supposed to have roles in tumor invasion, metastasis, drug resistance, and recurrence after therapy. CSC, tumor microenvironment (TME), extracellular matrix (ECM) and epithelial-mesenchymal transition (EMT) all have cross-link and are affected from the stimuli one to each other. TME can alter ECM, and ECM can induce EMT while EMT, TME and ECM have effects on generation of cancer stem cells/malignant phenotype and enable invasion and metastasis [62,63,64]. Therefore each of these processes may be a part of the solution to cure cancer.

Together with technological developments drug delivery systems (DDSs) have been highly improved recent years but there are still many challenges to face to proceed for clinical implementation for CSC-targeted DDSs [65].

In this part we focus on the factors that have effects on the therapy of OPC through targeting CSC properties.

### 4.1. CSC Plasticity

Because of CSC plasticity, heterogeneity is an obstacle of targeting CSC. Recognizing that cancer cells have the possibility to gain CSC abilities, both genetically and phenotypically the complexity of tumor heterogeneity highly increases [8]. Regarding the same type of cancer, such as, here, oropharyngeal cancer, different populations in the same tumor complicates to proceed to a solution for therapy. As previously mentioned, we discussed ALDH1A1 as a valuable prognostic marker in HNSCC but there are reports displaying contradictory results. For example, in one study, ALDH1A1 was found to be uniquely expressed in a subset of suprabasal tonsillar crypt epithelium and was lost in HPV^+^ and HPV^−^ tumors suggesting ALDH1A1 positive cells not to be stem cell progenitors but a component of the crypt cellular microenvironment [66]. Additionally Xu et al. showed that ALDH1A1 may be a biomarker for predicting lymph node metastasis, but it is not an independent prognostic factor for survival in HNSCC patients [67]. These conflicts may be explained by sub-classification of CSCs according to their expression of marker proteins which may have different positions/roles in cancer as suggested by Geißler et al. Their data revealed that the amounts of CD44 and ALDH1A1 vary; while ALDH1A1 high tumor cells express low levels of CD44 and EGFR, ALDH1A1^-^/CD44^+^ high tumor cells express high levels of EGFR in HNSCC. They suggested that CSCs can also be sub-classified into migratory and stationary CSCs. They proposed that ALDH1A1 high/CD44 low/EGFR low tumor cells may be stationary and quiescent, while ALDH1A1^−^/CD44 high/EGFR high cells may be invasive and migratory [68]. In a systematic review it was concluded that a single common CSC sorting marker may not even exist within identical types of tumor [69]. However, this raises the question instead of isolating CSCs with more than one marker would not be more enlightening to isolate CSCs with only one single marker but with the different CSC markers within the same sample to compare their roles in the carcinogenesis related processes, metastasis, resistance to therapy, therapy relapse tumor rather than grouping all CSC as one group. If there are different CSC markers in the identical types of tumor, do the CSCs expressing different markers have different roles in the carcinogenesis, invasion, metastasis, evading immune surveillance, resistance to therapy, therapy relapse? If there is more than one origin of CSC-like stem cells, progenitor cells and/or differentiated cells, is it possible that the origin of the CSC may cause differently expressed CSC markers in the same type of tumor? There are still a lot of questions to be answered. Many of the studies in HNSCC are focused on the prognosis of the patients comparing the expression of CSC markers or functional role of solely on one CSC marker. When differently expressed CSC markers and results are to be reported in the identical type of cancer in different research we can functionally and mechanistically (not only as the expression) compare the role of CSCs expressing different markers within the same tumor to have more precise results.

Not only for tumor heterogeneity but also for metastasis, CSCs are considered to have critical roles [70,71,72]. We do not know clearly yet whether a subgroup of CSCs metastasizes in OPC. Therefore studies revealing the role/function of different CSC subgroups are necessary for targeting CSC in OPC. These studies may have effect on preventing invasion, metastasis as well as curing OPC.

### 4.2. Resistance to Antitumor Therapies and Evade Immune Surveillance

CSCs are considered to be resistant to antitumor therapies and have the ability to escape immune surveillance [15,73,74,75]. For resistance to chemotherapy and radiation CSCs use mechanisms such as dormancy, DNA repair, multidrug-resistance-type membrane transporters, and escaping apoptosis [76]. Moreover, CSC markers such as CD44, ALDH1A1 have been reported to be intensely linked with EGFR and PI3K/AKT pathway. Coexistence of CD44v3 and ALDH1A1 in head and neck cancer cells provides escape from apoptosis, promotes survival and proliferation through activation of downstream effectors such as Sox2, Nanog, and Oct4 [77]. Not only resistance to therapeutics but recent therapies such as ionizing radiation therapy or cisplatin may even cause CSC characteristics of cancer cells is another challenging point [78,79]. Pützer et al. discussed that rather than unilateral anti-CSC approaches strengthening patient’s immune defense and heading toward individualized therapies CSC treatment can be successful [80].

### 4.3. CSC Niche

Because of the difficulties in the way of cure, targeting CSC alone may not be a solution to cure oropharyngeal cancer. For example, the aforementioned CSC plasticity and additionally cancer cells’ ability to gain CSC phenotype suggests complexity in targeting only one type of CSC. Additionally CSC niche, which is the tumor microenvironment in which CSC characteristics are regulated and supports self-renewal and survival of CSCs, can both be part of the problem and the solution. CD44 intracellular signaling in response to extracellular signals is reported as a mediator of the link between tumor-associated macrophages in the tumor microenvironment and CSCs [28]. Although we still do not know much about CSC niche, targeting CSC niche and crosstalk between CSCs may be an effective and durable way of cancer therapy [81]. Therefore studies that improve our understanding of the CSC niche are important for further developments.

### 4.4. CSC Mitochondria

CSC mitochondria are another target of the CSC research aiming at the therapy. In CSC, mitochondria have been shown to have a corresponding role like they serve in stem cells in the regulation of stem cell identity, differentiation and fate [82]. Not only these functions but also mitochondria can modify cell metabolism and can cause CSC evade apoptosis leading to survival of cancer cells as well [83]. In a clinical pilot study, CSCs were shown to be selectively eradicated through targeting mitochondria using doxycycline in early breast cancer patients [84]. There are also studies with promising results suggesting eradicating CSCs in cancer aiming mitochondria via antibiotics and vitamin C supplement [6,85,86].

## 5. Conclusions

After they were first defined in 1997 in leukemia, researchers around the world contributed to CSC research with valuable data that improved our understanding of CSCs. Now we can question more how to proceed for an effective therapy through CSC and CSC related factors. For HNSCC it has only been 14 years since CSCs were identified and accumulating information leads us to question the reason of controversial data. As we mentioned in prior parts these may be due to experimental related factors or may be due to different lineages and of their different role even in the same tumor type. Moreover, when considering OPC we not only face the same problems but lack of knowledge solely on OPC and different mechanisms for HPV-negative and HPV-positive cases. Therefore, there is still need for a vast number of further studies which would enlighten our understanding of CSC related characteristics and pathways of carcinogenesis, resistance to therapy and escaping immune surveillance, CSC plasticity, CSC niche and CSC mitochondria in OPC together with HPV status. To have an effective and a permanent therapy for OPC we believe all these factors have great importance and should be revealed. Targeting not only CSC but targeting cancer cells and CSC niche as well might be the preferable way to cure OPC (Figure 2).

## Figures and Tables

**Figure 1 cancers-13-03878-f001:**
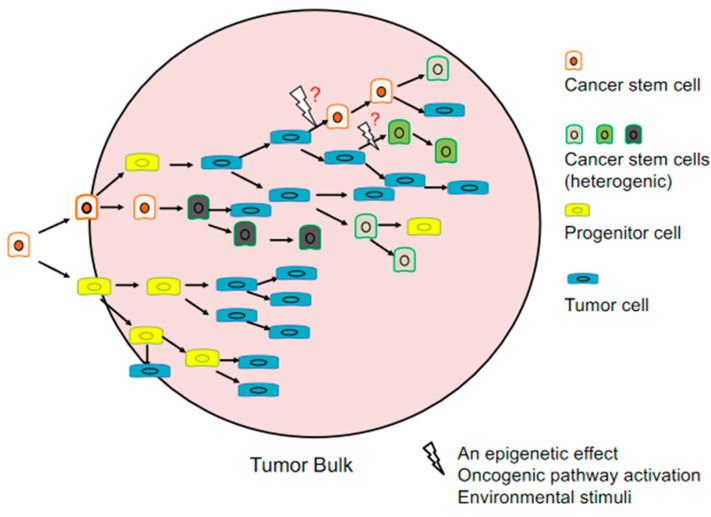
Origin and heterogeneity in cancer stem cell model; Cancer stem cells (CSCs) can form progenitor and cancer stem cells and can differentiate to cancer cells. When an appropriate epigenetic effect, oncogenic pathway activation or environmental stimuli event occur, differentiated cancer cells can transform to cancer stem cell.

**Figure 2 cancers-13-03878-f002:**
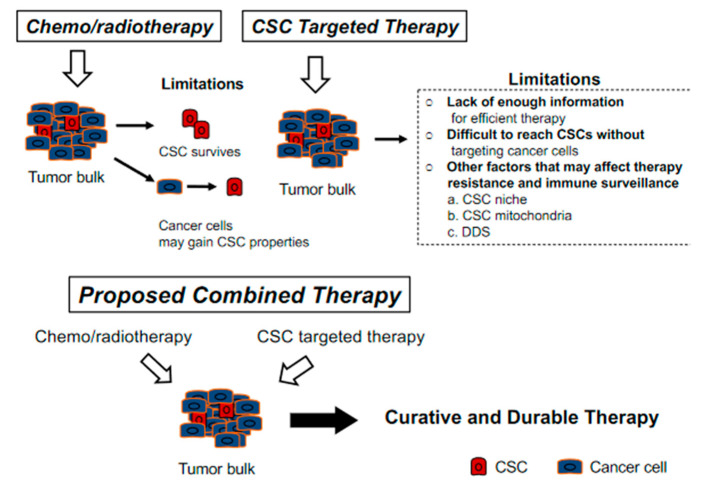
There are difficulties in each of conventional chemoradiotherapy and cancer stem cell (CSC) targeted therapy. While chemoradiotherapy leaves CSC untouched, CSC-targeted therapy leaves large tumor bulk with cancer cells, which may then gain CSC properties. Moreover, heterogeneity in various CSCs is another challenge to overcome.

**Table 1 cancers-13-03878-t001:** Differences between HPV-positive and negative oropharyngeal cancer *.

Subjects	HPV-Positive	HPV-Negative
Age	Younger	Elder
Smoking/Alcohol	Less	Often
Radiochemotherapy response	Well	High resistant rate
Survival	Better	Worse
Genetic alterations		
P53 Mutation	3%	84%
CDKN2A Mutation	none	58%
CCND1 Mutation	3%	31%
FGFR1	none	10%

* Genetic alterations data are from Cancer Genome Atlas Network [6].

**Table 2 cancers-13-03878-t002:** Common Cancer Stem Cell Markers in Oropharyngeal Cancer *.

Cancer Stem Cell Marker	Chromosomal Location	Exon Count	Characteristics
ALDH1A1	9q21.13	13	major pathway of alcohol metabolism, most common CSC marker
CD44	11p13	21	cell-surface glycoprotein involved in cell-cell interactions, cell adhesion and migration
BMI1	10p12.2	10	a proto-oncogene, a member of polycomb group complex 1 (PRC1) which is an epigenetic repressor of regulatory genes
OCT4	6p21.33	6	a transcription factor that plays role in embryonic development and stem cell pluripotency
SOX2	3q26.33	1(no introns)	SRY-box transcription factor 2
CD133	4p15.32	35	transmembrane glycoprotein expressed on adult stem cells, suppress differentiation to maintain stem cell properties

* https://www.ncbi.nlm.nih.gov/gene/ (accessed on 20 April 2021) data are used for Table 2.

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
