# Peer review of "Cancer Stem Cells in Oropharyngeal Cancer"

_cancers, 2021, doi:10.3390/cancers13153878_

Round 1

Reviewer 1 Report

A nice review of the topic.

There are a few areas where minor editing would make the paper more readable. For example line 225-226 - "Besides even cancer stem cells..." does not make sense - it might be reworded to something like "Recognizing that cancer stem cells....

Author Response

Reviewer 1

There are a few areas where minor editing would make the paper more readable. For example, line 225-226 - "Besides even cancer stem cells..." does not make sense - it might be reworded to something like "Recognizing that cancer stem cells.

Response to reviewer 1

Page 2: ‘undissolved’ is corrected as ‘unresolved’

Page 4: word ‘heterogenous’ is corrected as ‘heterogeneous’

Page 6: radioresistance is corrected as resistance to radiotherapy

Page 7: ‘worse’ is corrected as ‘and worse’

Page 11: ‘Besides even cancer cells have the possibility to gain CSC abilities that highly increase the complexity of tumor heterogeneity both genetically and phenotypically’ is corrected as ‘Recognizing that cancer cells have the possibility to gain CSC abilities, both genetically and phenotypically the complexity of tumor heterogeneity highly increases’

Page 13: ‘additional to this’is corrected as ‘additionally’

Reviewer 2 Report

The authors were aimed present CSC origin and model followed by CSC markers of HNSCC, reviewing the recent literature about CSC  of OPC, and limitations of our current knowledge. Moreover they discuss different perspectives, which may connect to better diagnostic as well as prognostic and therapeutic options, including targeted therapy for OPC.

The study covers some issues that have been overlooked in other similar topics. The study was conducted with a good scientifically sound. The structure of the manuscript appears adequate and well divided in the sub-paragraphs.

Issues that need improvement: Please check English grammar typos thorough the text.

Introduction section: Will be very useful for the readers to stress better the role of HPV molecular mechanism in the progression of the disease, taking in account some useful tools to detect the same  (please see and briefly discuss: please see and briefly discuss: https://doi.org/10.3390/v13040559;  doi:10.2174/1871530319666190119103255).

Conclusion Section: This paragraph required a general revision to eliminate redundant sentences and to add some "take-home" message.

Author Response

Reviewer 2:

  1. Issues that need improvement: Please check English grammar typos
    thorough the text.
  2. Introduction section: Will be very useful for the readers to stress
    better the role of HPV molecular mechanism in the progression of the
    disease, taking in account some useful tools to detect the same (please
    see and briefly discuss: please see and briefly discuss: https://doi.org/10.3390/v13040559; doi:10.2174/1871530319666190119103255).
  3. Conclusion Section: This paragraph required a general revision to eliminate redundant sentences and to add some "take-home" message.

Response to Reviewer 2

Point 1

Page 2: ‘undissolved’ is corrected as ‘unresolved’

Page 4: word ‘heterogenous’ is corrected as ‘heterogeneous’

Page 6: radioresistance is corrected as resistance to radiotherapy

Page 7: ‘worse’ is corrected as ‘and worse’

Page 11: ‘Besides even cancer cells have the possibility to gain CSC abilities that highly increase the complexity of tumor heterogeneity both genetically and phenotypically’ is corrected as ‘Recognizing that cancer cells have the possibility to gain CSC abilities, both genetically and phenotypically the complexity of tumor heterogeneity highly increases’

Page 13: ‘additional to this’is corrected as ‘additionally’

Point2

Following sentences are added as the reviewer suggested. Therefore 2 reference articles are added to the references as well.

Page 3: Mechanism of HPV integration to the host genome is not clear yet but fusions through break-points of cellular and viral genome or the amplified segments of a genomic sequence flanked with HPV genome which is also found in patient samples as focal copy number elevation at sites of HPV integration are mainly suggested mechanisms.

Page 4: There are also conflicting reports about HPV status and HNSCC which may be due to not screening cancer of the oropharynx and the anterior oral cavity separately or different sampling techniques likesaliva, biopsies, and brushing and methods used to detect HPV status from those samples through polymerase chain reaction (PCR), dot-blot hybridization, and Southern blotting.

Point3

Conclusion part has been revised and take-home message is added: After they were firstly defined in 1997 in leukemia researchers around the world contributed to CSC research with valuable data that improved our understanding about CSCs. Now we can question more how to proceed for an effective therapy through CSC and CSC related factors. For HNSCC it’s only been 14 years after CSCs were identified and accumulating information leads us to question the reason of controversial data. As we’ve mentioned in prior parts these may be due to experimental related factors or may be due to different lineages and of their different role even in the same tumor type. Moreover when considering OPC we not only face the same problems but lack of knowledge solely on OPC and different mechanisms for HPV negative and HPV positive cases. Therefore there is still need of a vast amount of further studies which would enlighten our understanding about CSC related characteristics and pathways of carcinogenesis, resistance to therapy and escaping immune surveillance, CSC plasticity, CSC niche and CSC mitochondria in OPC together with HPV status. To have an effective and a permanent therapy of OPC we believe all these factors have great importance and should be unrevealed. Targeting not only CSC but targeting cancer cells and CSC niche as well might be the preferable way to cure OPC (Figure 2).